# Serological Variety and Antimicrobial Resistance in *Salmonella* Isolated from Reptiles

**DOI:** 10.3390/biology11060836

**Published:** 2022-05-29

**Authors:** Lina Merkevičienė, Česlova Butrimaitė-Ambrozevičienė, Gerardas Paškevičius, Alma Pikūnienė, Marius Virgailis, Jurgita Dailidavičienė, Agila Daukšienė, Rita Šiugždinienė, Modestas Ruzauskas

**Affiliations:** 1Department of Anatomy and Physiology, Lithuanian University of Health Sciences, Tilžės g. 18, LT-47181 Kaunas, Lithuania; lina.merkeviciene@lsmuni.lt (L.M.); jurgita.dailidaviciene@lsmuni.lt (J.D.); agila.dauksiene@lsmuni.lt (A.D.); 2Department of Bacteriological Investigations, National Food and Veterinary Risk Assessment Institute, J. Kairiūkščio g. 10, LT-08409 Vilniu, Lithuania; ceslova.ambrozeviciene@nmvrvi.lt; 3Life Sciences Center, Vilnius University, Saulėtekio al. 7, LT-10257 Vilnius, Lithuania; gerardasp@yahoo.com; 4Lithuanian Zoological Garden, Radvilėnų pl. 21, 50299 Kaunas, Lithuania; alma.pikuniene@zoosodas.lt; 5Microbiology and Virology Institute, Lithuanian University of Health Sciences, Tilžės g. 18, LT-47181 Kaunas, Lithuania; marius.virgailis@lsmuni.lt (M.V.); rita.siugzdiniene@lsmuni.lt (R.Š.); 6Institute of Animal Rearing Technologies, Lithuanian University of Health Sciences, Tilžės g. 18, LT-47181 Kaunas, Lithuania

**Keywords:** antimicrobial susceptibility, epidemiology, lizards, reptiles, snakes, *Salmonella enterica*

## Abstract

**Simple Summary:**

Reptiles are carriers of different zoonotic pathogens hazardous to other animals and humans. *Salmonella* *enterica* is one of the best adapted bacterial pathogens causing infections. The aim of this study was to investigate the prevalence of *Salmonella* in different reptile species and to evaluate their serological variety and patterns of antimicrobial resistance. In total, 97 samples from 25 wild and domesticated reptile species were investigated in Lithuania for the presence of *Salmonella*. Fifty isolates of *Salmonella* were obtained from the ninety-seven tested samples. Results demonstrated that lizards and snakes are frequent carriers of a large variety of *Salmonella* serovars. Sixty-eight per cent of *Salmonella* were resistant to at least one antimicrobial. The most frequent resistance of the isolates was to streptomycin (26%), cefoxitin, gentamicin, tetracycline and chloramphenicol (16%). Genes encoding resistance to different antimicrobial classes were detected. The data obtained provided knowledge on *Salmonella* prevalence in reptiles. Healthy individuals, irrespective of their origin, often carry *Salmonella*, including multi-resistant strains. Due to its large serological diversity, zoonotic potential and antimicrobial resistance, *Salmonella* in reptiles poses a risk to other animals and humans.

**Abstract:**

*Salmonella* *enterica* is one of the best adapted bacterial pathogens causing infections in a wide variety of vertebrate species. The aim of this study was to investigate the prevalence of *Salmonella* in different reptile species and to evaluate their serological variety and patterns of antimicrobial resistance. In total, 97 samples from 25 wild and domesticated reptile species were investigated in Lithuania. Serological variety, as well as phenotypical and genotypical resistance to antimicrobials, were investigated. Fifty isolates of *Salmonella* were obtained from the ninety-seven tested samples (51.5%; 95% CI 41.2–61.2). A significantly higher prevalence of *Salmonella* was detected in domesticated individuals (61.3%; 95% CI 50.0–71.5) compared with wild ones (18.2%; 95% CI 7.3–38.5). All isolates belonged to a single species, *Salmonella enterica*. Results demonstrated that reptiles carry a large variety of *Salmonella* serovars. Thirty-four isolates (68%) of *Salmonella* were resistant to at least one antimicrobial drug. The most frequent resistance of the isolates was to streptomycin (26%), cefoxitin, gentamicin, tetracycline and chloramphenicol (16%). Genes encoding resistance to tetracyclines, aminoglycosides, sulphonamides and trimethoprim were detected. No integrons that are associated with horizontal gene transfer were found. Data obtained provided knowledge about the adaptation of *Salmonella* in reptiles. Healthy individuals, irrespective of their origin, often carry *Salmonella*, including multi-resistant strains. Due to its large serological diversity, zoonotic potential and antimicrobial resistance, *Salmonella* in reptiles poses a risk to other animals and humans.

## 1. Introduction

*Salmonella* is a well-known pathogen that is prevalent in multiple species of vertebrates. Although the carriage of *Salmonella* species (*Salmonella bongori* and *Salmonella enterica*) in the intestinal tract of reptiles usually does not cause illness to themselves, it can cause serious infections in people, in particular young children, elderly people or immunocompromised individuals. *Salmonella* is a zoonotic bacterium and was isolated from multiple vertebrate species, including both warm- and cold-blooded animals [1,2]. 

*Salmonella* is divided into 60 serogroups and more than 2300 serovars [3]. Except for characterizing clinical aspects of a few serovars, such as *Salmonella enterica* (*S*. *enterica*) serovar Typhi, serogrouping and serotyping are mainly used as public health tools to recognize outbreaks and identify and control sources of infection [3,4]. *Salmonellaenterica* spp. *enterica* serovars are considered zoonotic or potentially zoonotic. The most common serovars infecting humans worldwide are *S*. serovar (ser.) Typhimurium and *S*. ser. Enteritidis [1].

Humans may become infected through direct contact with reptiles or indirectly by manipulating objects, home stuff or contaminated food [1,5,6,7,8,9]. Globally, it is estimated that there are 93.8 million cases of salmonellosis per year caused by different reasons [10,11]. It is estimated that over 70,000 people get salmonellosis from reptiles each year in the United States, while 160,649 human cases of salmonellosis were reported in 2006 in 25 European countries, including Bulgaria, Romania, Iceland, Liechtenstein, Latvia, Germany, France and Norway [12,13].

Humans and animals share bacterial species including resistant ones. Antibiotic resistance of bacteria to antimicrobials is currently a primary concern in both human and veterinary medicine. For this reason, epidemiological studies in domestic and wild animals should be performed on a regular basis [1,2]. Resistant pathogens, including *Salmonella enterica*, should be of particular attention as these bacteria are very well adapted to different hosts, carry different genes encoding for both virulence and antimicrobial resistance and are currently among the most common infectious agents isolated from humans with food-borne infections. The aim of this study was to investigate the prevalence of *Salmonella* in different reptile species and to evaluate their serological variety and patterns of antimicrobial resistance. 

## 2. Materials and Methods

### 2.1. Samples and Place 

In 2020–2021, samples (n = 97) of domesticated reptile faeces and cloacal swabs of wild reptiles were collected using sterile cotton swabs with a transport medium (Transwab^®^ Amies, Corsham, UK). Domesticated reptiles such as pet animals were sampled all over Lithuania from private keepers as well as in the Lithuanian Zoo. All animals were clinically healthy and underwent physical examination by a veterinarian before sampling. No treatments with antibiotics were performed for at least 6 months before sampling. Wild reptiles were caught and samples were collected from three main locations in Lithuania: Raguva (55.56472 24.61574); (55.564476 24.617858); the Rumšiškės forest (54.881027 24.178046); (54.881832 24.178994) and Čepkeliai—Dzūkija National Park (54.0214 24.4289), (54.0225 24.4831), (54.0562 24.4241), (54.0606 24.4302). Ethical approval for this study was given by the Lithuanian Environmental Protection Agency (permissions numbers AS-4800 and AS-4884).

Samples were delivered to the laboratory within 24 h of collection, kept in containers with transport media on ice for 1–2 h then followed by refrigeration at +2–4 °C. In total, 97 samples were collected from 25 different species of reptiles (Table 1). 

### 2.2. Isolation and Identification of Salmonella

Isolation of *Salmonella* was performed according to the EN ISO 6579-1 (ISO, 2017) [14] procedure for *Salmonella* detection. Xylose lysine deoxycholate (XLD) agar and *Salmonella Shigella* (SS) agar (Oxoid, Basingstoke, UK) were used as plating media after the enrichment procedure. The randomly selected separate colonies (one colony per sample) were identified using the “Microgen Gram-Negative Plus” biochemical identification system (Microgen, Camberley, UK). 

*Salmonella* serotyping was carried out by standard slide agglutination test (CEN ISO/TR 6579-3:2014) [15] with polyvalent and monovalent somatic (O) and flagella (H) antisera (Statens Serum Institute Denmark and Sifin, Berlin, Germany). Firstly, suspected colonies were picked up and tested with somatic O polyvalent and O polyvalent group antisera. In the case of a positive reaction, testing according to the Kaufman–White scheme was applied. If the suspect colony did not show any reaction with O polyvalent antisera, *Salmonella* species and subspecies were identified by biochemical properties. Serotyping results were evaluated according to the Kaufmann–White *Salmonella* serotyping scheme.

### 2.3. Susceptibility Testing

Antimicrobial susceptibility testing was performed using the disk diffusion method according to Kirby-Bauer. Antimicrobials of different classes were selected with the aim of addressing the risk of salmonellosis to public health. The following disks were used: ampicillin (10), cefoxitin (30), gentamicin (10), chloramphenicol (30), sulfamethoxazole-trimethoprim (25), cefpodoxime (10), ciprofloxacin (5), tetracycline (30), ofloxacin (5), streptomycin (10) and doxycycline (30). The results were interpreted according to the European Committee on Antimicrobial Susceptibility Testing licensed by EUCAST 2022 clinical breakpoints [16] whenever possible. For tetracycline, doxycycline, cefpodoxime and streptomycin the interpretation of the results was performed using Clinical and Laboratory Standards Institute (CLSI) guidelines [17]. In the case of resistance to at least three or more antimicrobial classes, the isolates were treated as multi-resistant isolates.

### 2.4. Molecular Testing

The resistant *Salmonella* isolates were tested by polymerase chain reaction (PCR) for detection of the genes encoding resistance. DNA material for molecular testing was obtained after bacterial lysis was performed as described previously [18]. PCR included 30 cycles of denaturation (94 °C, 30 s), annealing (30 s) and extension (94 °C, 90 s). Annealing temperatures and oligonucleotides used are presented in Table 2. As a negative control, DNA/RNA-free water was used instead of the antigen whereas, for the positive control strains, Enterobacteriaceae from the culture collection of the Microbiology and Virology Institute at the Lithuanian University of Health Sciences were used.

### 2.5. Data Analysis

Statistical analysis was performed using the IBM SPSS Statistics package, version 27 (SPSS Inc., Chicago, IL, USA). For percentage estimates, Wilson (score) 95% confidence intervals (CI 95%) and their ranges for true population proportions were calculated. Comparison between categorical variables was calculated using a chi-squared test or Fisher’s exact test for small counts. Results were considered statistically significant if *p* < 0.05. The number of genes encoding resistance to separate antimicrobials was expressed in % from the number of resistant isolates tested. 

## 3. Results

### 3.1. Salmonella Prevalence in Reptiles

In total, 97 reptile samples were tested, of which 22 samples came from wild reptiles and 75 samples from domesticated animals. Fifty animals were positive for *Salmonella* (51.5%; 95% CI 41.2–61.2), as determined by isolation of the cultures with further biochemical identification. All of the isolates belonged to a single species, *Salmonella enterica*. The results demonstrated that the frequency of *Salmonella* prevalence was significantly higher in domesticated reptiles than in wild ones (*p* < 0.0475). Forty-six isolates (61.3% 95% CI 50.0–71.5) were obtained from domesticated reptiles and four (18.2%; 95% CI 7.3–38.5) were obtained from wild individuals. Overall, *Salmonella* was isolated from 17 out of the 25 reptile species (68%) included in this study. The prevalence of *Salmonella* in different reptile species is presented in Table 3.

### 3.2. Serological Variety of the Salmonella Isolates

In total, 34 *Salmonella* isolates showing antimicrobial resistance were serotyped. Twenty-seven out of thirty-four (79.4%) isolates had a positive reaction only with O (somatic) antisera, whereas only three strains had a positive reaction with H grouping antisera. Different serogroups and serovars were identified including IIIa, enterica arizonae/IIIb, enterica diarizonae, Sherbrooke, Maiduguri, Waycross, Macallen, and others. Most of the isolates belonged to O:4, O:8 and O:18 serogroups; some serogroups, including O:41, O:30 and O:3.10 were only detected in up to three isolates, and some were detected just by single isolates. Characteristics of *Salmonella* isolated from domesticated and wild reptiles according to their serological patterns are presented in Table 4 and Table 5. 

### 3.3. Antimicrobial Resistance

Of 50 *Salmonella* isolates, 34 (68%) were resistant to at least one tested antimicrobial. The resistant isolates recovered from domesticated and wild reptiles are presented in Table 6.

In total, 24 of 50 isolates were resistant (48%) to a single or two antimicrobial agents, whereas 10 (20%) of the isolates were multi-resistant, i.e., resistant to three or more antimicrobial classes. Multi-resistant isolates were obtained from grass snakes (*Natrix natrix*) (n = 3), boa constrictors (*Boa constrictor*) (n = 2), a Chinese water dragon (*Physignathus cocincinus*) (n = 1), a California kingsnake (*Lampropeltis californiae*) (n = 1), a corn snake (*Pantherophis guttatus*) (n = 1), a milk snake (*Lampropeltis triangulum*) (n = 1) and a king ratsnake (*Elaphe carinata carinata*) (n = 1). The phenotypical resistance of the isolates is presented in Figure 1.

The data demonstrated that the frequency of the resistance to different antimicrobials was not high; however, the spectrum of resistance was wide, i.e., there was no antimicrobial substance tested that was effective against all *Salmonella* isolates. The most frequent resistance prevalence of the isolates was against streptomycin (26%; chi-squared test, *p* < 0.00073), cefoxitin, gentamicin, tetracycline and chloramphenicol (16%; chi-squared test, *p* < 0.001). 

The genes encoding antimicrobial resistance to different antibiotics were as follows: *aadA* (37.5%) and *armA* (37.5%) encoding resistance to aminoglycosides, *sul*2 (50%) encoding resistance to sulphonamides, *dfr*1 (50%) and *dfr*7 (16.6%) encoding resistance to trimethoprim and *tet*A (61.5%) and *tet*B (53.8%) encoding resistance to tetracyclines. No genes were detected for encoding the resistance to β-lactams, fluoroquinolones, and amphenicols. No integrons associated with horizontal gene transfer were detected. The data of susceptibility profiles as well as the genes encoding resistance are presented in Table 4 and Table 5.

## 4. Discussion

*Salmone**lla* is a gram-negative pathogen that causes various host-specific diseases. It is one of the most widespread agents causing human gastrointestinal infections, as well as infections in pigs, poultry and calves. Although the clinical significance of *Salmonella* infections in wild and captive reptiles is poorly understood, it is thought that the majority of infections lead to an asymptomatic carrier state and do not result in disease [1]. In this study, all tested reptiles were clinically healthy, therefore our data support this opinion. From the reptiles sampled in this study, *Salmonella* was found in 61% and 18% of domesticated and wild individuals, respectively, across Lithuania. In other studies, the data were quite similar. For example, in central Europe (Poland, Germany and Austria), the prevalence of *Salmonella* in domesticated snakes and lizards ranged from 33% to 54.1% [44]. In Norwegian zoos, *Salmonella* was recovered from 62% of snakes and 67% of lizards [1]. Although there is a lack of data about *Salmonella* prevalence in wild reptiles, a study performed by Scheelings et al. demonstrated a higher prevalence of this bacterium in reptiles held in captivity (47%) compared to wild reptiles (14%) [2]. Such data are very similar to the data obtained in our study; however, the number of wild animals used in our study was low. More wild animals should be investigated in order to answer whether domesticated animals more often carry *Salmonella* than wild individuals. The origin of *Salmonella* in both captive and wild reptiles is also unclear. As some of the wild species, such as *Coronella austriaca*, dwell far away from the urban areas, it may be assumed that the carriage of *Salmonella* in reptiles is not necessarily associated with human activity, but this microorganism can be a part of the natural microbiota of reptiles. On the other hand, a higher prevalence of *Salmonella* in domesticated reptiles rather than in wild individuals, as detected in this study, may be explained by the mixing of different reptile species in a single premise, restricted area, and carriage by humans. Feed intended for reptiles can also be a reason for *Salmonella* spread because either raw feed (such as live or frozen rodents) or concentrated feed may be contaminated by *Salmonella*. 

In this study, the most frequent carriage of *Salmonella* was detected in snakes, especially in *Lampropeltis californiae*, *Pantherophis guttatus*, *Lampropeltis triangulum* and *Elaphe taeniura friesei* and less frequently in lizards. This may be associated with a smaller number of investigated lizards, as in other studies, *Salmonella* was more frequently isolated from lizards rather than from snakes [1,44]. This may also depend on investigated species of lizards, as different species of lizards both in the wild and in captivity have different feed diets; some lizards eat arthropods and other invertebrates while larger species include small vertebrates in their diet [45]. As rodents are known as reservoirs of different pathogens including *Salmonella*, this fact of *Salmonella* epidemiology in reptiles could be considered very important and should be further studied.

The exact serotyping of *Salmonella* in reptile isolates is not always successful because of a wide variety of serovars and that the most well-known serovars with epidemiological importance for humans and domestic animals are less frequently presented in cold-blooded animals. Different serovars of *Salmonella* were detected in this study, including *Salmonella* Florida, *Salmonella* Sherbrooke, *Salmonella* Maiduguri, *Salmonella* Waycross and others, whereas the most widespread serovars in humans and farm animals according to responsible institutions and previous data in Lithuania were *Salmonella* Enteritidis, *Salmonella* Typhimurium, *Salmonella* Choleraesuis, *Salmonella* Infantis, *Salmonella* Derby and *Salmonella* Dublin [46]. Although studies about *Salmonella* serotypes in reptiles are scarce, some data from other countries exist. For instance, in Norway, 26 different *Salmonella* serovars in captive reptiles were detected, including those with high zoonotic potential, such as *Salmonella* Paratyphi B, subsp. *arizonae*, and those with low or moderate zoonotic potential, such as serovar *Salmonella* Lome, subsp. *salamae*, *diarizonae* and others [1]. In neighbouring Poland, 209 serovars of *Salmonella* were detected in reptiles, from which the most prevalent were *Salmonella* Oranienburg, *Salmonella* Tennessee, *Salmonella* Agona, *Salmonella* Fluntern and *Salmonella* Muenchen [47]. In French Guiana, 14 different *Salmonella* serovars were detected among wild reptiles. Interestingly, nearly two-thirds of the *Salmonella* serovars isolated from reptiles were also isolated from patients in this country [48]. The high prevalence of *Salmonella* in humans and overlapping serotypes was explained by the handling and consumption of reptiles by humans. Such data support the fact that *Salmonella*, regardless of the serovars, is pathogenic and can easily be transmitted from reptiles to humans. 

The data on antimicrobial susceptibility of the isolates revealed a wide spectrum of resistance, as there were no antimicrobials tested that would be effective for all *Salmonella* isolates. This can be explained by the large diversity of *Salmonella* among reptiles. In a recent study performed in Poland, *Salmonella* isolates from reptiles were most frequently resistant to streptomycin [47], which is in accordance with the results obtained in our study. In Taiwan, the most frequent resistances were to streptomycin and tetracycline [8]. In Spain, the most frequent resistances among *Salmonella* from reptiles were to gentamicin, colistin, and ampicillin [49]. In this study, 20% of all *Salmonella* isolates were multi-resistant. Such strains usually pose a high risk not only for the treatment of infections but also for the transfer of resistance genes to other microbiota. Horizontal gene transfer is very common among Enterobacteriaceae including *Salmonella*; however, we did not detect integrons that are associated with horizontal gene transfer. Nevertheless, as *Salmonella* can be easily transferred to humans or other animals and are pathogenic, multi-resistant strains are of great concern. Different studies demonstrate the unequal frequency of multi-resistant *Salmonella* isolated from reptiles. For example, in Poland, only single multi-resistant strains were isolated [47], whereas, in Spain, 72% of the isolates were multi-resistant [49]. Such data also prove the large diversity of *Salmonella* among reptiles. Although we have detected some genes encoding antimicrobial resistance, their variety was not high, especially when compared with isolates from farm animals or humans. Genes encoding resistance to tetracyclines, aminoglycosides, sulphonamides and trimethoprim were detected. Although the same genes were recently detected in Enterobacteriaceae from domestic (unpublished data) and wild animals in Lithuania [50], it is difficult to make any conclusion about their origin in *Salmonella* isolates from reptiles. Further studies are needed to analyse possible relations of microorganism transfer between reptiles and other hosts. 

## 5. Conclusions

Reptiles are carriers of a wide variety of serovars and multi-resistant strains of *Salmonella*. Although the prevalence of *Salmonella* was higher in domesticated reptiles than in wild individuals, further studies are needed to support this theory, as the number of tested wild animals was much lower than in domesticated ones. Reptiles can be a reservoir of *Salmonella*, therefore hygienic measures should be kept when maintaining and carrying them, as well as when keeping reptiles in close contact with other animals. 

## Figures and Tables

**Figure 1 biology-11-00836-f001:**
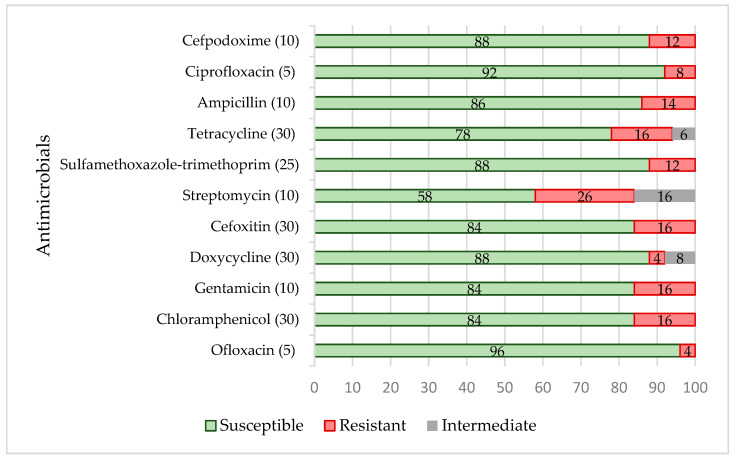
Phenotypical antimicrobial resistance (%) patterns of the *Salmonella* isolates from reptiles (n = 50). Intermediate describes the zone of inhibition in between “susceptible” and “resistant”. The numbers in brackets near the antimicrobial agent represent the antimicrobial concentrations (µg) of the discs.

**Table 1 biology-11-00836-t001:** Species and number of tested reptiles.

Domesticated and Wild Reptile Species	
Snakes	Number
Grass snake (*Natrix natrix*)	13
California kingsnake (*Lampropeltis californiae*)	9
King ratsnake (*Elaphe carinata carinata*)	7
Taiwan beauty ratsnake (*Elaphe taeniura friesei*)	6
Mexican vine snake (*Oxybelis aeneus*)	6
Corn snake (*Pantherophis guttatus*)	5
Milk snake (*Lampropeltis triangulum*)	5
Desert kingsnake (*Lampropeltis splendida*)	5
Smooth snake (*Coronella austriaca*)	4
Boa constrictor (*Boa constrictor*)	3
Brown house snake (*Boaedon capensis*)	2
Ball python (*Python regius*)	2
Western hognose snake (*Heterodon nasicus*)	1
Banded water snake (*Nerodia fasciata*)	1
Total:	69
Lizards	Number
Slow worm (*Anguis fragilis*)	5
Central bearded dragon (*Pogona vitticeps*)	5
Crested gecko (*Correlophus ciliatus*)	4
Great plated lizard (*Gerrhosaurus major*)	3
Frill-necked lizard (*Chlamydosaurus kingii*)	2
Plumed basilisk (*Basiliscus plumifrons*)	2
Chinese water dragon (*Physignathus cocincinus*)	2
Common chameleon (*Chamaeleo chamaeleon*)	1
Green iguana (*Iguana iguana*)	1
Common leopard gecko (*Eublepharis macularius*)	1
Total:	26
Turtles	Number
Central Asian tortoise (*Testudo horsfieldii*)	2
Total:	2

**Table 2 biology-11-00836-t002:** Antimicrobial resistance genes tested and oligonucleotide primers used in the study.

Primer Name	Sequence (5′-3′)	Size, bp and t (°C)	Target Gene	Source
*bla*TEM-F	GAGTATTCAACATTTTCGT	857 (50)	*tem*	[19]
*bla*TEM-R	ACCAATGCTTAATCAGTGA			
*bla*SHV-F	TCGCCTGTGTATTATCTCCC	768 (60)	*shv*	[20]
*bla*SHV-R	CGCAGATAAATCACCACAATG			
*oxa*1-F	TCAACAAATCGCCAGAGAAG	276 (55)	*OXA* group I	[21]
*oxa*1-R	TCCCACACCAGAAAAACCAG			
*oxa*3-F	TTTTCTGTTGTTTGGGTTTT	427 (52)	*OXA* group III	
*oxa*3-R	TTTCTTGGCTTTTATGCTTG			
*OXA* 5 group-F	AGCCGCATATTTAGTTCTAG	644 (56)	*OXA* group V	
*OXA* 5 group-R	ACCTCAGTTCCTTTCTCTAC			
*CTX*-M-F	ATGTGCAGYACCAGTAARGT	593 (50)	*ctxM*	[22]
*CTX-*M-R	TGGGTRAARTARGTSACCAGA			
*cmy*2-F	GCACTTAGCCACCTATACGGCAG	758 (58)	*cmy*	[23]
*cmy*2-R	GCTTTTCAAGAATGCGCCAGG			
*PER*-1-F	ATGAATGTCATTATAAAAGCT	927 (48)	*per*	[24]
*PER*-1-R	TTAATTTGGGCTTAGGG			
*PER*-2-F	ATGAATGTCATCACAAAATG	927 (49)		
*PER*-2-R	TCAATCCGGACTCACT			
*tet*A-F	GTGAAACCCAACATACCCC	888 (55)	*tet*A	[25]
*tet*A-R	GAAGGCAAGCAGGATGTAG			
*tet*B-F	CCTTATCATGCCAGTCTTGC	774 (55)	*tet*B	
*tet*B-R	ACTGCCGTTTTTTCGCC			
aadB-F	ATGGACACAACGCAGGTCGC	534 (55)	*aad*B	[26]
aadB-R	TTAGGCCGCATATCGCGACC			
*aad*A-F	GTGGATGGCGGCCTGAAGCC	528 (68)	*aad*A	
*aad*A-R	AATGCCCAGTCGGCAGCG			
*rmt*B-F	ATGAACATCAACGATGCCCT	769 (55)	*rmt*B	[27]
*rmt*B-R	CCTTCTGATTGGCTTATCCA			
*arm*A-F	CAAATGGATAAGAATGATGTT	774 (55)	*arm*A	[28]
*arm*A-R	TTATTTCTGAAATCCACT			
*aph*A1-F	AAACGTCTTGCTCGAGGC	500 (55)	*aph*A1	[29]
*aphA*1-R	CAAACCGTTATTCATTCGTGA			
*aacA*4-F	ATGACTGAGCATGACCTTGCG	487 (55)	*aacA4*	[30]
*aacA*4-R	TTAGGCATCACTGCGTGTTCG			
*aac*(3)II-F	TGAAACGCTGACGGAGCCTC	369 (65)	*aac*(*3*)*II*	[31]
*aac*(3)II-R	GTCGAACAG GTAGCACTGAG			
*str*A-F	CCTGGTGATAACGGCAATTC	546 (55)	*str*A	[32]
*str*A-R	CCAATCGCAGATAGAAGGC			
*str*B-F	ATCGTCAAGGGATTGAAACC	509 (55)	*str*B	
*str*B-R	GGATCGTAGAACATATTGGC			
*cat*II-F	ACACTTTGCCCTTTATCGTC	495 (55)	*cat*II	[33]
*cat*II-R	TGAAAGCCATCACATACTGC			
*cm*lA-F	TTGCAACAGTACGTGACAT	293 (55)	*cml*A	[34]
*cml*A-R	ACACAACGTGTACAACCAG			
*sul*1-F	TTCGGCATTCTGAATCTCAC	822 (55)	*sul*1-F	[35]
*sul*1-R	ATGATCTAACCCTCGGTCTC			
*sul*2-F	CGGCATCGTCAACATAACC	722 (50)	*sul*2-F	[36]
*sul*2-R	GTGTGCGGATGAAGTCAG			
*sul*3-F	GAGCAAGATTTTTGGAATCG	792 (51)	*sul*3-F	
*sul*3-R	CATCTGCAGCTAACCTAGGGCTTTGA			
*Dfr1*-F	ACGGATCCTGGCTGTTGGTTGGACGC	254 (55)	*dfr*1	[37]
*Dfr1*-R	CGGAATTCACCTTCCGGCTCGATGTC			
*Dfr5*-F	GCBAAAGGDGARCAGCT	394 (44)	*dfr*5	[38]
*Dfr5*-R	TTTMCCAYATTTGATAGC			
*DfrA7*-F	AAAATTTCATTGATTTCTGCA	471 (44)	*dfr*7	[39]
*DfrA7*-R	TTAGCCTTTTTTCCAAATCT			
*qnrA-F*	ATTTCTCACGCCAGGATTTG	516 (53)	qnrA	[40]
*qnrA-R*	GATCGGCAAAGGTTAGGTCA			
*qnrB-F*	GATCGTGAAAGCCAGAAAGG	469 (53)	qnrB	
*qnrB-R*	ACGATGCCTGGTAGTTGTCC			
*qnrS-F*	ACGACATTCGTCAACTGCAA	417(53)	qnrS	
*qnrS-R*	TAAATTGGCACCCTGTAGGC			
*qepA-F*	CAGTGGACATAAGCCTGTTC	218 (60)	qepA	[41]
*qepA-R*	CCCGAGGCATAGACTGTA			
*teg1-F*	TTATTGCTGGGATTAGGC	164 (55)	integrase I class	[42]
*teg1-R*	ACGGCTACCCTCTGTTATC			
*teg2-F*	ACGACATTCGTCAACTGCAA	233 (50)	integrase II class	[43]
*teg2-R*	TAAATTGGCACCCTGTAGGC			

**Table 3 biology-11-00836-t003:** Species of reptiles carrying *Salmonella* isolates.

Domesticated Reptile Species	Number of *Salmonella* Carriers/Tested
California kingsnake (*Lampropeltis californiae*)	8 of 9
Desert kingsnake (*Lampropeltis splendida*)	5 of 5
Mexican vine snake (*Oxybelis aeneus*)	5 of 6
Taiwan beauty rat snake (*Elaphe taeniura friesei*)	5 of 6
Corn snake (*Pantherophis guttatus*)	4 of 5
Milk snake (*Lampropeltis triangulum*)	4 of 5
King ratsnake (*Elaphe carinata carinata*)	4 of 7
Crested gecko (*Correlophus ciliatus*)	2 of 4
Boa constrictor (*Boa constrictor*)	2 of 3
Brown house snake (*Boaedon capensis*)	2 of 2
Western hognose snake (*Heterodon nasicus*)	1 of 1
Great plated lizard (*Gerrhosaurus major*)	1 of 3
Banded water snake (*Nerodia fasciata*)	1 of 1
Common leopard gecko (*Eublepharis macularius*)	1 of 1
Chinese water dragon (*Physignathus cocincinus*)	1 of 2
Total:	46 of 60
Wild reptile species	Number
Grass snake (*Natrix natrix*)	3 of 13
Smooth snake (*Coronella austriaca*)	1 of 4
Total:	4 of 17

**Table 4 biology-11-00836-t004:** Characteristics of *Salmonella* isolated from domesticated reptiles.

Reptile Species	Identification by Biochemical Testing	Phenotypic Resistance	Genotypic Resistance	Serovar or Serogroup
Boa constrictor (*Boa constrictor*)	*Salmonella* sub.2	TE, CN, PX	*tetA*	IIIa, enterica arizonae; IIIb enterica diarizonae
Milk snake (*Lampropeltis triangulum*)	*Salmonella* sub.3 A *S*. *enterica* subsp. *arizonae*	TE, CN, C, PX	*aadA*,	IIIa, enterica arizonae; IIIb enterica diarizonae
Boa constrictor (*Boa constrictor*)	*Salmonella* sub.4	TE, CN, C, STR, AMP, CIP	*aadA*,	O:8
Brown house snake (*Boaedon capensis*)	*Salmonella* sub.3B	CN, STR	*aadA*,	Florida
Taiwan beauty rat snake (*Elaphe taeniura friesei*)	*Salmonella* sub.5	CN	*-*	O:4
Corn snake (*Pantherophis guttatus*)	*Salmonella* sub.3 A *S*. *enterica* subsp. *arizonae*	CN	*armA*	O:65
King ratsnake (*Elaphe carinata carinata*)	*Salmonella* sub.2	FOX, CN, STR, PX	*armA*	O:4
Crested gecko (*Correlophus ciliatus*)	*Salmonella* sub.3 A *S*. *enterica* subsp. *arizonae*	FOX, C	-	Sherbrooke
Crested gecko (*Correlophus ciliatus*)	*Salmonella* sub.3 A *S*. *enterica* subsp. *arizonae*	STR	-	Maiduguri
Desert kingsnake (*Lampropeltis splendida*)	*Salmonella* sub.I A *S*. *enterica*	FOX, TE,	*tetA*, *tet B*	O:41
Mexican vine snake (*Oxybelis aeneus*)	*Salmonella* sub.3 A *S*. *enterica* subsp. *arizonae*	STR	-	Waycross
California kingsnake (*Lampropeltis californiae*)	*Salmonella* sub.1 *S*. *enterica*	DO, TE,	*tetA*, *tet B*	Waycross
California kingsnake (*Lampropeltis californiae*)	*S*. *enterica* subsp. *S bongori* V	STX, STR	*dfr1*	O48, IIIa/IIIb
Corn snake (*Pantherophis guttatus*)	*Salmonella* group 2 *S*. *enterica* subsp. *arizonae*	CN, SXT, STR, AMP	*armA*, *sul2*, *dfr1*	O:4
Corn snake (*Pantherophis guttatus*)	*Salmonella* group2 *S*. *enterica* subsp. *arizonae*	STR	-	O44 IIIa/IV
Milk snake (*Lampropeltis triangulum*)	*Salmonella* sub.3 A *S*. *enterica subsp*. *arizonae*	STR	-	O:57
Western hognose snake (*Heterodon nasicus*)	*Salmonella* sub.3 A *S*. *enterica subsp*. *arizonae*	STR	-	O:65 IIIb
Great plated lizard (*Gerrhosaurus major*)	*Salmonella* sub.2	C	-	O:41
California kingsnake (*Lampropeltis californiae*)	*Salmonella* sub.2	STR	-	O:8
Banded water snake (*Nerodia fasciata*)	*Salmonella* sub.3 A *S*. *enterica subsp*.*arizonae*	STR	-	O:50
Milk snake (*Lampropeltis triangulum*)	*Salmonella* sub.2	AMP	-	O:30
Common leopard gecko (*Eublepharis macularius*)	*Salmonella* sub.2	STR	-	O:18
Taiwan beauty rat snake (*Elaphe taeniura friesei*)	*Salmonella* sub.3 A *S*. *enterica subsp*. *arizonae*	STR	-	O:18 (K)
Brown house snake (*Boaedon capensis*)	*Salmonella* sub.3 A *S*. *enterica subsp*. *arizonae*	C	-	O:8 (C2–C3)
Desert kingsnake (*Lampropeltis splendida*)	*Salmonella* sub.2	CIP	-	O:3,10 (E1)
King ratsnake (*Elaphe carinata carinata*)	*Salmonella* sub.3 A *S*. *enterica subsp*. *arizonae*	TE	*tetA*, *tet B*	O:3.10 (E1)
California kingsnake (*Lampropeltis californiae*)	*Salmonella* sub.3 A *S*. *enterica subsp*. *arizonae*	FOX, TE, SXT, PX, AMP, OFX	*tetA*, *tetB*	O:1; 3.19 (E)
Taiwan beauty rat snake (*Elaphe taeniura friesei*)	*Salmonella* sub.3 A *S*. *enterica subsp*. *arizonae*	FOX, CIP,	*-*	O:40 (R)
Mexican vine snake (*Oxybelis aeneus*)	*Salmonella* sub.3 A *S*. *enterica subsp*. *arizonae*	TE	*tetA*, *tetB*	O:44 (V)
Chinese water dragon (*Physignathus cocincinus*)	*Salmonella* sub.3 A *S*. *enterica subsp*. *arizonae*	FOX, TE, SXT, C, STR, PX, AMP	*sul2*, *dfr7*	O:30

AMP: ampicillin; FOX: cefoxitin; STR: streptomycin; CN: gentamicin; TE: tetracycline; DO: doxycycline; C: chloramphenicol; CIP: ciprofloxacin; OFX: ofloxacin; SXT: sulfamethoxazole-trimethoprim; PX: cefpodoxime.

**Table 5 biology-11-00836-t005:** Characteristics of *Salmonella* isolated from wild reptiles.

Reptile Species	Identification by Biochemical Testing	Phenotypic Resistance	Genotypic Resistance	Serovar or Serogroup
Smooth snake (*Coronella austriaca*)	*Salmonella* group I	TE, STR	*tetA*, *tetB*	O:43 (U)
Grass snake (*Natrix natrix*)	*Salmonella* sub.3 A *S*. *enterica subsp*. *arizonae*	DO, FOX, TE, SXT, STR, PX, AMP	*dfr1*	O:3,10 (E1), Macallen
Grass snake (*Natrix natrix*)	*Salmonella* sub.2	FOX, TE, C, STR, AMP	*tetA*, *tetB*	O:18
Grass snake (*Natrix natrix*)	*Salmonella* sub.2	TE, SXT, AMP, CIP, OFX	*sul2*	O:18 (K)

AMP: ampicillin; FOX: cefoxitin; STR: streptomycin; TE: tetracycline; DO: doxycycline; C: chloramphenicol; CIP: ciprofloxacin; OFX: ofloxacin; SXT: sulfamethoxazole-trimethoprim; PX: cefpodoxime.

**Table 6 biology-11-00836-t006:** Species of reptiles carrying antimicrobial resistant *Salmonella* isolates.

Domesticated Reptile Species	Number
California kingsnake (*Lampropeltis californiae*)	4
Corn snake (*Pantherophis guttatus*)	3
Milk snake (*Lampropeltis triangulum*)	3
Taiwan beauty rat snake (*Elaphe taeniura friesei*)	3
Desert kingsnake (*Lampropeltis splendida*)	2
Mexican vine snake (*Oxybelis aeneus*)	2
King ratsnake (*Elaphe carinata carinata*)	2
Crested gecko (*Correlophus ciliatus*)	2
Boa constrictor (*Boa constrictor*)	2
Brown house snake (*Boaedon capensis*)	2
Western hognose snake (*Heterodon nasicus*)	1
Great plated lizard (*Gerrhosaurus major*)	1
Banded water snake (*Nerodia fasciata*)	1
Common leopard gecko (*Eublepharis macularius*)	1
Chinese water dragon (*Physignathus cocincinus*)	1
Total:	30
Wild reptile species	Number
Grass snake (*Natrix natrix*)	3
Smooth snake (*Coronella austriaca*)	1
Total:	4

## Data Availability

Not applicable.

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
