# Peer review of "Serological Variety and Antimicrobial Resistance in *Salmonella* Isolated from Reptiles"

_biology, 2022, doi:10.3390/biology11060836_

Round 1
Reviewer 1 Report
The current study reports the prevalence of Salmonella spp from wild and domestic reptiles in the region of Lithuania and assessed their sensitivity to wide range of antibiotics generally used in treating salmonellosis. Authors report that domestic reptiles have higher rate of positivity over the wild animals. About 70% of the isolates show resistance to at least one of the of the commonly used antibiotics. Authors have also tested the presence of resistance gene towards the respective antibiotics in the isolates.
The current study has scientific merits mainly due to the isolation and characterization of wild isolates; however, the conclusions drawn need more supporting evidences. Firstly, the claim that the domestic reptiles are predominant Salmonella carriers, has to be re-evaluated. The number of wild host samples are limiting in comparison to the wide range of domestic hosts analyzed. Though the previous epidemiological studies show similar trend, the current data set needs more wild host samples to substantiate the claim or the authors have to reword the statement. Secondly, for identification of resistance gene section, please share more details about the PCR, what are the controls used? Was the amplicon sequenced? Lastly, and more importantly, the manuscript is full of grammatical and typographic errors. It has to be extensively edited.
Author Response
Authors are very thankful for the remarks and for the possibility to improve our manuscript. Please find attached response to the remarks.

Reviewer 2 Report
Dear Editor,
thank you for considering me as reviewer. I have carefully read the manuscript and I’ve found it very interesting. In my opinion, the manuscript deserves publication, after some minor revisions.
Here I provide my comments and suggestions.
KEYWORDS
It is possible to add “reptiles”?
ABS
- L44: antimicrobial drug
KEYWORDS
- It is possible to add “reptiles”?
INTRODUCTION
- L61: change “>” with “more than”
- L 64: please, state that “S” is the abbreviation of Salmonella the first time that you mention this word and then maintain the abbreviation along the text. Please, do not use abbreviation at the beginning of a sentence.
- L66: “ser” à same as “S” (look at my previous comment.
- L 67-72: References are missing. Moreover, it is not clear the aim of this paragraph considering the first paragraph of the introduction. Please, consider rephrasing in order o be clearer to readers or to remove this part.
- L74: manipulate objects instead of place hands on
- L74-76: considering the following rephrasing “Humans may become infected through direct contact with reptiles or indirectly, manipulating objects, home stuff or contaminated food”
- L81: Antibiotic resistance of bacteria …
MATERIAL AND METHODS
- It is important to explain to readers if animals were healthy or not (if not, explain symptoms and signs). I can deduce some information reading the results, but it is important to state this information here in order to give strength to the study design. Moreover, explain if the animals received a physical/ clinical examination by a veterinarian before sampling. Therefore, it is important to know if antimicrobial drugs have been administered to reptiles in the two weeks prior the sampling: this could be a bias for the further susceptibility tests and the hypothesis about AMR.
- L104: How samples have been stored in this 24h- period?
- L118-119: This sentences is a results and you have inserted it correctly in the results section. Please, remove this sentence here.
- L135: identified instead of determinated
- L136: were evaluated
- L 140, section 2.3: please, explain why you chose these compounds. I do not think it’s a reason about the clinical purpose (otherwise you should select enrofloxacin or marbofloxacin instead of ciprofloxacin).
- L145: licensed by EUCAST 2022
- L147:guidelines
- Explain what “slight modifications” are.
- L179: double spaced “six ..isolates »
- L185 : Font 9. Also for Table 3
- L195: de What Authors mean with “up to 3 cases”?
- Table 3: Integrons in a single line, please
- L210: “,” after isolates
- L217: according to your statement about MDR strains, you have to describe to readers the criteria that you considered to establish if a stain was MDR or not. For example: resistant to more than 3 molecules or resistant to more than 3 families of antimicrobial drugs. This is a key point. Please, provide also pertinent reference/references (these two classification criteria are equally true but based on different considerations that have been widely discussed in literature).
- Fig1 caption: font 9
- L 228: different instead of separate
- L231: against instead of was demonstrated to
- L238: fluroquinolones instead of (fluoro)quinolones since you just considered ciprofloxacin
- L246: add a reference at the end of the sentence
- L246-249: This argument is not well explained (the terms used are not clear and English is poor, here). Reference is missing at the end of the sentence.
- L252-253: that’s why it is important to explain to readers before if animals were checked.
- L253: considering that the instead of as (the first, at the beginning of the sentence)
- L254: , after asymptomatic
- L259: ranged from … to (remove was)
- L266: “live” : I didn’t get the meaning of this word here…sorry…could you rephrase in order to be clearer?
- L296: , after on the other hand
- L281: Reference is missing after diet.
- L297-299: Why Authors have considered Australia? The climate situation and the reptiles species present there are completely different to Lithuania. Please, explain to readers the reason behind this citation.
- L299:In the neighboring Poland,
- L302: why “i.e.”? In which reptiles these bacteria have been identified in Poland?
- L306: such data support the fact that instead of confirm
- L334-335: too rough! ? Please, explain better
L400-401: check why these two lines are empty (probably editing/layout oversight)
Author Response
Authors are very thankful for your time and for providing valuable remarks that were very useful for improving of our manuscript. Please find our responses to your remarks on separate file.

Reviewer 3 Report
First of all, I would like to thank the authors for their work on this research and manuscript. Overall, the paper is very good with regards to the scientific content and how it has been written.
The majority of my comments relate to some very minor grammar and punctuation errors, so should be easy to address. There are a handful of more significant comments however, that I hope the authors take time to address.
- Line 179 - Please report the p-value. "p<0.05" is unsuitable for publication. The test statistic is also not reported.
- The authors are using a point (.) when writing large numbers. E.g. for twenty-thousand they are writing 20.000 whilst I appreciate that is normal in many countries, the scientific convention is to use commas e.g. 20,000. Given that, elsewhere, they have used "." as a decimal place (e.g. line 76 "93.8 million"), they need to change to be consistent.
- Line 77: 70.000 70,000
- Line 78: 160.649 160,649
- Line 30 - missing commas, change to: Healthy individuals, irrespective of their origin, often carry Salmonella including multi-resistant 30
strains. - Line 34-35, missing word: causing infections in a
wide variety - Line 41 - in comparing compared with wild ones
- Line 58 - Salmonella is a zoonotic bacteria
- Line 67-69 - Salmonella is generally considered a normal constituent of reptilian intestinal microbiota although there are not much studies, particularly in separate species of reptiles there are few studies investigating this, particularly with regards to the extent of the bacterium across different reptile species.
- Line 75 - of reptiles or in their mouths
- Line 82-83 - is one of the main problem a primary concern in both both in human and veterinary medicine
- Line 83 - commas missing: By this reason, epidemiological
- Line 84-85 - commas missing: Resistant pathogens, including
S.almonella enterica, should - Line 95 - after the word "samples" it's best to include: (n=97)
- Line 104 - laboratory during within 24 hours of collection
- Line 105-117 - this list is very difficult to read. I would suggest authors use some form of table to make it easier and more concise. Perhaps, for each reptile group (e.g. snakes, lizards) two columns: species and n.
- Table 2 - Reporting the number of individuals carrying Salmonella is not that useful without the total number in the same table. For the second column I suggest writing something like: 8/12 or 8 (of 12) so the reader see how it compares to the total.
- Section 3.2 and 3.3 - These are confusing. In line 189 you state that "In total 34 Salmonella isolates showing antimicrobial resistance were serotyped." This is confusing to the reader because where did that number 34 come from? Later, in section 3.3 you explain that. This is the wrong way around. I think the order of section 3.2 and 3.3 need to be swapped.
- Figure 1 has great information in, but it isn't displayed particularly well.
- Figure 1 - you define a category "intermediate" however this is not defined anywhere. Please define what an "intermediate" result is.
- Figure 1 - I understand why you only included the samples which were resistant to at least one antimicrobial. However, this risks being misleading. It would have been better to include all samples as it would give a far better indication of scale to the reader, or at least provide reference to it in some other manner in the figure.
- Figure 1 - it is unclear what the number in brackets after the drug (e.g. Doxycycline (30)) means. The figure caption is poor at explaining the Figure.
- Line 232-233 - no test statistics given.
- Line 244-252 - that's a lot of unnecessary information for the discussion, most of that can be removed or included in the introduction. You need to get to the point quicker.
- Line 253-254 - As the popularity of reptiles as pets are is increasing and
- This is quite a definitive claim, if possible you need to provide a source for this information. I am not convinced it is true, without a source/citation.
- Line 256-257 - This study shows that reptiles carry Salmonella in 61% and 18% in domesticated and wild individuals respectively. This statement is false. You cannot possibly claim that your study is sufficient to accurately predict overall Salmonella rates at a larger level. You need to rewrite this to something like: From the reptiles sampled in this study Salmonella was found in 61% and 18% in domesticated and wild individuals (respectively) across Lithuania.
- Line 274 - Salmonella were was detected in snakes
- Line 309 - isolates revealed a wide spectrum of
- Line 312 - usually are delivered derived from
- Line 318 - In a recent study performed
- Line 312-346 - this is very long and wordy, especially for a discussion section.
- Line 354-355 - Wide variety of Salmonella serovars including rare detectable as well as epidemiologically important/zoonotic ones exists among reptiles. This sentence is poorly written and needs re-writing.
Author Response

(The authors gave the same response as above.)

Round 2
Reviewer 1 Report
The additional information, both in the discussion and description of the methodology, addresses my original concern with reasonable clarity. The language still needs to be edited.
Author Response
Dear Reviewer,
We have repeatedly sent the manuscript to the editing service and checked it by ourselves. We hope that the manuscript now is suitable for publication.
Thank you very much for your time and efforts.
On behalf of authors,
Modestas Ruzauskas